# AraC Functional Suppressors of Mutations in the C-Terminal Domain of the RpoA Subunit of the *Escherichia coli* RNA Polymerase

**DOI:** 10.3390/microorganisms12091928

**Published:** 2024-09-23

**Authors:** Dominique Belin, Jordan Costafrolaz, Filo Silva

**Affiliations:** 1Department of Pathology and Immunology, Faculty of Medicine, University of Geneva, 1 Rue Michel-Servet, CH-1211 Geneva, Switzerland; filomena.silva@unige.ch; 2Department of Microbiology and Molecular Medicine, Faculty of Medicine, University of Geneva, 1 Rue Michel-Servet, CH-1211 Geneva, Switzerland; jordan.costafrolaz@unige.ch

**Keywords:** arabinose operon, genetic suppression, transcription activation, P_BAD_ promoter

## Abstract

In *E. coli*, transcriptional activation is often mediated by the C-terminal domain of RpoA, the α subunit of RNA polymerase. Random mutations that prevent activation of the arabinose P_BAD_ promoter are clustered in the RpoA C-terminal domain (α-CTD). We have isolated functional suppressors of *rpoA* α-CTD mutations that map to *araC*, the main transcriptional regulator of *ara* genes, or to the P_BAD_ promoter. No mutation was found in the DNA regulatory region between *araC* and P_BAD_. Most suppressors that improve P_BAD_ transcription are localized to the N-terminal domain of AraC. One class of *araC* mutations generates substitutions in the core of the N-terminal domain, suggesting that they affect its conformation. Other suppressors localize to the flexible N-terminal arm of AraC. Some, but not all, suppressors confer an arabinose constitutive phenotype. Suppression by both classes of *araC* mutations requires the α-CTD to stimulate expression from P_BAD_. Surprisingly, in *rpoA*^+^ strains lacking Crp, the cAMP receptor protein, these *araC* mutations largely restore arabinose gene expression and can essentially bypass Crp activation. Thus, the N-terminal domain of AraC exhibits at least three distinct activities: dimerization, arabinose binding, and transcriptional activation. Finally, one mutation maps to the AraC C-terminal domain and can synergize with AraC mutations in the N-terminal domain.

## 1. Introduction

The arabinose regulon was the first system in which positive control of gene expression was genetically demonstrated [1]. AraC, the product of the activator gene, binds to arabinose and stimulates transcription of the catabolic genes *araB*, *araA*, and *araD* (*araBAD* operon), as well as those of the arabinose import systems, *araE* and *araFGH*. In the absence of arabinose, AraC represses transcription from the P_BAD_ promoter. In both instances, one AraC monomer is bound to the *I1* site (Figure 1). The transition between the repressor and activator states of AraC involves a large conformational movement, named the light switch, that displaces one subunit of the AraC dimer from the distant *O2* operator site to the promoter proximal *I2* activator site. During the switch, the N-terminal arm of AraC covers the arabinose-binding pocket of the N-terminal dimerization domain (AraC-NTD) [2,3,4]. The P_BAD_ promoter is intrinsically very weak, and AraC bound to the *I2* site enhances RNA polymerase binding and open complex formation [5]. In addition, Crp, the cAMP receptor protein, is required for full activity of the P_BAD_ promoter [6,7]. Crp activates many bacterial promoters, and several mechanisms of activation have been described [8]. The properties of the arabinose system were exploited to construct a widely used set of expression plasmids [9].

The α subunit of RNA polymerase, encoded by the *rpoA* gene, plays at least two roles in transcription. Its N-terminal domain (α-NTD) is involved in the assembly of the enzyme [10,11]. In contrast, the C-terminal domain (α-CTD), which is attached by a flexible linker to the α-NTD, plays multiple roles in transcription activation [8,12,13,14]. It activates transcription by interacting with activators in different ways and/or by binding to UP DNA sites upstream of promoters [15,16]. Since RNA polymerase contains two α subunits, each subunit can interact independently with two activating regions [13]. In addition to α-CTD structures [17,18], several structures of α-CTD-containing complexes have been resolved, including that of α-CTD bound to Fis at the *proP* P2 promoter [19], that of the α-CTD–Crp–DNA complex [14], and that of the RNA polymerase holoenzyme bound to a synthetic promoter in the presence of Crp [20].

At the arabinose P_BAD_ promoter, one α-CTD interacts with Crp [7]. The potential target of the second α-CTD has not yet been identified. However, by analogy with other Crp-dependent promoters [8], it has been proposed to bind the AraC DNA-binding domain and possibly a DNA motif localized between the Crp and AraC binding sites [3,21] (Figure 1). Several random *rpoA* α-CTD mutations that interfere with P_BAD_ activity have been characterized. 

We have used a genetic approach to search for functional suppressors that compensate for the detrimental effects of mutations in the RpoA α-CTD. Random mutagenesis of *araC* and of the intergenic regulatory region between *araC* and *araB* led to the isolation of *araC* suppressors. Most of these random mutations are scattered at several positions in the AraC-NTD, and their effect requires the RpoA α-CTD. We have also characterized a mutation in the DNA-binding C-terminal domain of AraC that exerts a synergistic effect with mutations in the AraC-NTD. However, we did not identify an AraC contact site with the RpoA α-CTD. Rather, all the mutations may promote or stabilize a conformation of AraC that more efficiently promotes transcription, thereby functionally compensating for the deleterious effects of the *rpoA* mutations.

## 2. Experimental Procedures

**Strains.** MC4100 is an F-*araD*139 Δ(*argF-lac*)U169 *flhD*5301 *fruA*25 *relA*1 *rpsL*150 *rbsR*22 Δ(*fimB-fimE*)632 *deoC*1 *thi* strain [22]. DB504 is MC4100 *ara*Δ*714 malE18* [23]. JW5702 contains the Δ*crp*-*765*::*kan* allele [24] and was obtained from the Coli Genetic Stock Center (CGSC strain 11596). HfrH was obtained from the Coli Genetic Stock Center (CGSC strain 5081). 

The *rpoA* mutations were introduced into DB504 by bacteriophage P1-mediated co-transduction with a *gspA*::Tn*10* marker (90–95% co-transduction) [25]. The presence of the *rpoA* mutations was verified by suppression of the toxic phenotype caused by induction of the appropriate proteins, by growth on M63 minimal media with or without cysteine and methionine, and by fermentation on McConkey-melibiose indicator plates [26]. The Δ*crp*-765::*kan* allele was introduced by P1-mediated transduction in strains carrying the *rpoA* alleles; the transductants were plated on streptomycin (100 µg/mL) containing plates to select for recombination between the *rpsL150* marker of DB504 and *crp*, and retention of the *gspA*::Tn*10* marker. Strains containing the Δ*crp*-*765*::*kan* allele were grown in LB containing 0.1% glucose [25]. Information on the *E. coli* genes and regulatory sequences was obtained from https://ecocyc.org/ (accessed on 25 August 2024).

**Plasmids.** Plasmids derived from pBAD24 [9] contain the *araC* gene, the intergenic region between *araC* and P_BAD_, the P_BAD_ promoter, and the first 13 nt of the *araBAD* transcript. pBAD18-vs.1 expresses a lysozyme encoded by bacteriophage T4 [27]. pBAD72K expresses a chimeric protein containing the signal sequence of murine PAI2 fused to the mature portion of PhoA [28]. pDB2114 expresses gene *55.2* of bacteriophage T4 [29]. These three plasmids confer a toxic phenotype that prevents growth in the presence of arabinose. 

Maspin (SERPINB5, P70124) is a member of the ovalbumin-related protease inhibitors family [30] whose N-terminal region is devoid of signal sequence activity [31]. psM74c expresses a Maspin::PhoA chimera: the initial Met is fused to residues 23–46 of Maspin, with a P32L substitution to allow export, followed by a synthetic ALA-A c-region cleaved by signal peptidase, and residues 23–471 of PhoA (P00634). The export of this chimeric protein is slightly more efficient than that of wild-type PhoA; hence psM74c was used to isolate and characterize suppressors of the *rpoA* mutations. 

The *rpoA*^+^ gene was PCR amplified from chromosomal DNA with the primers RPOAUP (5′cgggatcccacctgatcgtcg) and RPOADON (5′ccggtaccttaacctgtgatccggttactcg), and cloned between the BamHI and KpnI restriction sites of pUC19 to generate pUC-*rpoA*. The deletion of the α-CTD was performed by PCR amplification of pUC-*rpoA* with the primers RPOAUP and RPOAdel (5′cggggtaccttacacttcaggctgacg). The pUC-*rpoA*del plasmid expresses amino acids 1 to 242 of RpoA. The expression level of the proteins from the P_lac_ promoter is estimated to be ~50-fold higher than that from the chromosomal *rpoA* gene in *rpoA*^+^ *crp*^+^ strains; the level should be ~5-fold in Δ*crp* strains. The pUC-*rpoA*del plasmid confers a Mel^−^ phenotype in an *rpoA*^+^ strain.

The original pBAD24-derived plasmids (AraC^+^, AraC D7N, and AraC I71V) were digested with NsiI and XbaI. The *araC*-Maspin::PhoA fragments were cloned into the cognate sites of pBAD33 whose origin is compatible with that of the *rpoA* plasmids [9]. 

A complete deletion of the *araC* coding region was performed by removing the two MfeI-digested fragments of psM74c.

**Enzyme assays.** PhoA activity was qualitatively detected on LB plates supplemented with 0.2% arabinose and 40 µg/mL XP (5-bromo-4-chloro-3 indolyl phosphate). PhoA enzymatic activity was determined by measuring the rate of *p*-nitro-phenyl-phosphate hydrolysis [32]. Cultures were assayed after a 1 h induction with 0.2% arabinose. LacZ activity was determined by measuring the rate of *o*-nitro-phenyl-galactoside hydrolysis [25]. For each strain, triplicate independent cultures were assayed twice. The values were analysed with a Mann–Whitney test or an unpaired *t*-test.

**Random mutagenesis.** Hydroxylamine mutagenesis of psM74c plasmid DNA was performed as described [31]. Upon transformation in DB504, approximately 6% of the colonies carried inactivating mutations in *araC*, the regulatory region, or the maspin::*phoA* gene and formed white colonies on PhoA indicator plates containing arabinose. Plasmids allowing the formation of light blue colonies in *rpoA* Δ*crp* strains occurred at a frequency of 1–2%; in these strains, the parental plasmid formed white colonies. More than half of the mutant plasmids exhibited a higher plasmid copy number than the original plasmid. The complete *araC* gene was recloned in the parental plasmid. In a second experiment, a complete *araC* fragment (NsiI-MluI, 1130 bp long) was randomly mutagenized by PCR [28] and cloned in psM74c. Ten percent of the resulting plasmids transformed into an *rpoA*^+^ *crp*^+^ strain formed white colonies on PhoA indicator plates containing arabinose. After transformation in the *rpoA* A272E Δ*crp* strain, candidate suppressors were sequenced. Three additional AraC residues were isolated: N16S, I51T, and I71V; three mutants had different substitutions at previously identified residues (S14P, A152T, and A152P), and the A152V substitution was isolated again. The *araC* mutations described here have been independently isolated once (D7N, N16S, I51T, I71V, A152T, and A152P), or 4–6 times (S14L, S14P, G22D, and A152V). To isolate suppressors of *rpoA* N268T, the mutagenized *araC* PCR fragment was digested with NsiI and EcoRV and cloned in psM74c. After transformation in the *rpoA* N268T *crp*^+^ strain, three suppressors were identified: R61H was isolated six times, while R261C and C205Y were isolated twice.

## 3. Results

### 3.1. RpoA Mutations in the α-CTD Region That Interfere with P_BAD_ Expression

The chimeric protein PAI2::PhoA contains the signal sequence of murine PAI-2 fused to the mature portion of PhoA. Expression of this chimeric protein from the arabinose P_BAD_ promoter is lethal because it interferes with protein export to the periplasm [31]. Suppressors of this toxicity were selected as colony-forming mutants in the presence of arabinose. While many chromosomal suppressors map to *sec* genes involved in protein export to the periplasm thus preventing the toxicity of the chimera [28,33], we also isolated fifteen *rpoA* mutants (Table 1). To modulate the severity of the toxicity in our selective system, we used several genes of bacteriophage T4 that also confer a toxic phenotype when expressed from the arabinose P_BAD_ promoter. The expression of one of these genes, *vs.1*, encoding an exported lysozyme whose toxicity is weaker than that of PAI2::PhoA [27] led to the isolation of six *rpoA* mutants. Finally, one *rpoA* mutant was isolated as a suppressor of phage T4 gene *55.2*, a topoisomerase inhibitor that exhibit the strongest toxicity of the three proteins [29]. A total of twenty-two independent *rpoA* mutations have been isolated (Table 1). 

The randomly selected *rpoA* mutations all map to amino acids 268–272 of the RpoA C-terminal domain (α-CTD) (Figure 2A). The RpoA K271E substitution (*rpoA341* [26,34]) was the most frequent, but three other new substitutions (V, T, and I) were also identified at this position (Table 1). The four substitutions of K271 exert similar effects on P_BAD_ promoter expression, although the K271E substitution has a slightly stronger effect (Figure 2B, Table 1). These mutations affect CysB regulatory function and prevent growth on M63 minimal medium in the absence of cysteine or methionine (Cym phenotype) [26]. The L270F substitution appears to be the weakest of the *rpoA* alleles since it allows growth on minimal media (Table 1) and thus does not detectably interfere with activation of the *cysA* operon by CysB [35]. The N268T substitution is the strongest *rpoA* mutation observed (Figure 2B), as it prevents growth on minimal media, even in the presence of casamino acids. Thus, this mutation must affect one or more promoter(s) in addition to P_BAD_ and P*_cys_*_A_.

The *rpoA* mutations were selected on arabinose-containing LB plates, under conditions where the expression of the indicated toxic protein prevents growth. For each substitution, the number of independent isolates is shown. The N268T allele suppresses the toxicity of all three proteins. The L270F allele suppresses the toxicity of T4 *vs.1* and partially that of PAI2::PhoA. The other alleles suppress the toxicity of T4 *vs.1* and PAI2::PhoA but not that of T4 *55.2*.

Colonies grown on McConkey-melibiose indicator plates were scored as R (red), identical to those of *rpoA** strains; K (pink), reflecting an intermediate melibiose fermentation; and W (white), reflecting no detectable melibiose fermentation. On M63-minimal-glucose plates, only the weak L270F allele allows growth. Upon addition of Cys and Met, most *rpoA* mutants grew. The N268T allele must prevent the expression of one or more auxotrophic genes.

### 3.2. Suppressors of the rpoA Mutations

Both AraC and Crp promote transcription from the P_BAD_ promoter in the presence of arabinose [3,4]. While AraC is absolutely required, a 20- to 40-fold reduction in expression can be detected in an *rpoA*^+^ strain in the absence of Crp (Figure 3A). Quantification of *phoA* expression from the P_BAD_ promoter in strains deleted for *crp* and carrying five alleles of *rpoA* demonstrated a strong reduction of promoter activity with the four substitutions of K271, while the N268T mutation totally abolishes expression (Figure 3B). These findings imply that the α-CTD is required for P_BAD_ transcription in the absence of Crp. 

To genetically define whether the interactions defective in the RpoA α-CTD mutants can be functionally suppressed, we carried out a suppressor screen. We used Δ*crp* strains containing either the K271E or K271V RpoA substitution. We also employed a Maspin::PhoA fusion that contains a very efficient signal sequence. This chimeric protein provides an extremely sensitive reporter, compensating for the reduced expression from the P_BAD_ promoter in the absence of Crp. Cells were transformed with mutagenized plasmids harboring the *araC* gene and the P_BAD_ regulatory region and plated on LB plates containing arabinose and XP (a color indicator of PhoA activity). Colonies in which AraC-dependent expression of maspin*::phoA* is restored are blue. Plasmids isolated from such strains were sequenced. 

We isolated four plasmids that contain mutations in the P_BAD_ promoter (Figure 4). One of these, at position −36, was located in the −35 hexamer, between the *cip*-5 site [6,36] and the *I*^c^ mutations [37,38]. The *cip*-5 and *I^c^* promoters allow AraC-independent constitutive expression, and the −36 mutation could also facilitate RNA polymerase binding. The second one, at position −23, has a weaker stimulatory effect. The last two, located at position −10, are identical to the X^c^ mutation, which has been described as creating a strong AraC-independent promoter [36,39]; a P_BAD_ promoter carrying the X^c^ mutation is not affected by the *rpoA341* K271E mutation [26]. No mutation was isolated in the regulatory region between the *araC* coding sequence and the P_BAD_ promoter (Figure 1).

We isolated four mutations in *araC*. These *araC* suppressors all localize to the N-terminal domain of AraC (AraC-NTD). Since most *araC* constitutive mutations map to this region [41,42], we measured expression from the P_BAD_ promoter in the presence and absence of arabinose with these AraC mutant proteins. In a *rpoA*^+^ Δ*crp* strain, the A152V substitution confers a nearly complete constitutive phenotype (Figure 5A). In contrast, G22D and S14L confer an expression level of only 12–27% in the absence of the inducer (Figure 5A,C). Constitutive expression with A152V has been observed in a *crp*^+^ strain [42]. Finally, the D7N substitution is entirely devoid of constitutive activity (Figure 5C). In the *rpoA* K271E strain, only A152V confers a partial constitutive phenotype (34% of the induced level) (Figure 5B,D); this suggests that the RpoA α-CTD may stabilize the activator form of the constitutive AraC proteins. However, an arabinose constitutive phenotype is not necessary for functional suppression of the *rpoA* mutations.

While induced expression mediated by wild-type AraC is strongly decreased in an *rpoA*^+^ Δ*crp* strain, the *araC* mutations confer robust expression in this strain, at levels of 30–50% of that measured in an *araC*^+^ *rpoA*^+^ *crp*^+^ strain. Thus, they can partially compensate for the absence of Crp in *rpoA*^+^ strains (Figure 5A,C). 

These *araC* mutants were isolated in strains carrying either the K271E or K271V RpoA substitutions. We determined the capacity of the *araC* mutations to suppress the four different K271X substitutions, as well as the N268T one. The results shown in Figure 6 were obtained with AraC D7N and A152V; similar results were obtained with S14L and G22D (Appendix A). In induced *rpoA*^+^ *crp*^+^ strains, expression from the P_BAD_ promoter was slightly increased (1.5- to 3-fold) by the *araC* mutations (Figure 6A,C). In almost all cases, the expression observed with the *rpoA* mutants did not reach the levels observed in *rpoA*^+^ strains, indicating that the AraC suppressors do not fully compensate for the *rpoA* defect. However, the AraC D7N substitution restored the Ara^S^ phenotype conferred by PAI2::PhoA in an *rpoA* K271T *crp*^+^ strain; an effect of the other *araC* mutations could not be tested because of their constitutive activity. In Δ*crp* strains, stimulation by the *araC* mutations was much higher (Figure 6B,D). In *rpoA*^+^ Δ*crp* strains, the suppressors nearly compensated for the absence of Crp, as shown above in Figure 5. With the four *rpoA* K271 substitutions, the *araC* mutations increased expression 8- to 30-fold when compared to *araC*^+^. In no case was expression as high as that detected in *rpoA*^+^ strains, confirming that the *araC* mutations only partially compensate for the *rpoA* defects. Finally, the RpoA N268T substitution, which exerts the most drastic effect on P_BAD_ expression, was not suppressed by any of these *araC* mutations, both in the presence and absence of Crp, suggesting two independent mechanisms for α-CTD activation.

To determine whether any specificity could be observed between the AraC suppressors and other α-CTD mutated residues, we compared the K271E substitution with substitutions at the two flanking residues, L270 and A272 (Figure 7). The three mutant AraC proteins exerted similar effects in all cases, although small quantitative differences were detected. Expression was highest with the *rpoA* L270F mutant, the weakest *rpoA* allele, and lowest with the *rpoA* A272E one. When compared to *araC*^+^, increased expression with the *araC* mutants was found to be stronger in Δ*crp* (Figure 7) than in *crp*^+^ strains (Appendix A). In conclusion, we did not succeed in isolating a single *araC* mutation that selectively suppressed, in an allele-specific manner, any of the RpoA mutated residues tested.

In a different approach, using a mutagenic PCR protocol on a plasmid harboring the wild-type *araC* gene, we identified three additional AraC residues (N16S, I51T, and I71V) whose substitutions resulted in increased P_BAD_ expression in an *rpoA* A272E Δ*crp* strain. In addition, different substitutions were found at two previously identified residues (S14P and A152T). In the *rpoA*^+^ Δ*crp* strain, all compensated to a varying extent for the absence of Crp (Figure 8). The effect was maximal for the S14P mutant and minimal for the I51T mutant; the A152T mutant was less efficient than the A152V one (see Figure 6D). Furthermore, while the S14P and N16S mutants were partially constitutive, the other three mutants remained fully inducible; the A152T mutant, therefore, behaves clearly differently from the A152V mutant (Figure 8). These results confirm that a constitutive phenotype is not required for the suppression of the *rpoA* defects. The AraC I71V substitution restored the Ara^S^ phenotype conferred by PAI2::PhoA in the *rpoA* K271T *crp*^+^ strain, confirming that increased expression from the P_BAD_ promoter can also be observed with this mutant in the presence of Crp. In Δ*crp* mutant strains carrying different *rpoA* alleles, increased expression was also detected (Appendix A). Here again, we did not observe any allele specificity.

### 3.3. The Activity of the AraC Mutant Proteins Requires the α-CTD of RpoA

The observed functional suppression of RpoA α-CTD mutations and the lack of allele specificity of the AraC mutations are compatible with at least two models. Firstly, the suppressors could create or improve an interaction between the AraC-NTD and the RNA polymerase core. Alternatively, RpoA α-CTD could contact a specific site of AraC that remains to be identified. In this first scenario, functional suppression would not depend on the RpoA α-CTD. To discriminate between these two possibilities, we expressed either wild-type RpoA or an RpoA deletion that lacks the α-CTD (residues 243–329). Both proteins were expressed from the P_lac_ promoter on a plasmid, with levels predicted to be ~5-fold higher than that of RpoA derived from the chromosomal *rpoA*^+^ gene in the absence of Crp. 

In *rpoA*^+^ Δ*crp* strains, the activity of AraC^+^ decreased about 2-fold in the presence of the RpoA deletion (Figure 9A). A similar decrease was observed with the two AraC suppressors, namely the D7N mutant located in the AraC N-terminal arm and the I71V mutant located in the core of the AraC-NTD (Figure 9B,C). This result is more evident when the activity of the three AraC proteins in cells carrying the RpoA^+^ (Figure 9E) or the RpoA deletion plasmid (Figure 9F) are directly compared. As a control, expression from the *lacZ* promoter, which is activated only by an interaction of the RpoA α-CTD with Crp [8], was decreased to the same extent in the presence of the RpoA deletion (Figure 9D). When RpoA and the RpoA deletion were expressed from the strong bacteriophage T4 gene *32* promoter [43], we observed an almost complete inhibition of P_BAD_ expression (Appendix A). Taken together, these results demonstrate that the RpoA α-CTD is required for transcriptional activation at the P_BAD_ promoter with both wild-type and mutant AraC in the absence of Crp, suggesting that the AraC mutants functionally compensate for defects in the RpoA α-CTD by increasing AraC activation of RNA polymerase.

### 3.4. A Suppressor of RpoA N268T Synergizes with Mutations in the NTD of AraC

Since none of the AraC mutants described above suppressed the *rpoA* N268T mutant, even in the presence of Crp, we repeated the mutant screen in the *rpoA* N268T *crp*^+^ strain and isolated three mutants in the C-terminal DNA-binding domain of AraC. The largest effect was observed with R251H, while a 3-fold lower effect was observed with R251C and C205Y. The AraC R251H mutant compensated to the same extent for the effect of the *rpoA* N268T mutant in a *crp*^+^ strain (Figure 10A) or in the absence of Crp in a *rpoA*^+^ strain (Figure 10B); in *rpoA* K271X *crp*^+^ strains, expression was >50% of that measured in an *araC*^+^ *rpoA*^+^ *crp*^+^ strain. In contrast, this AraC mutant was totally inactive in the *rpoA* N268T Δ*crp* strain (Figure 10C).

Since R251H also strongly increased expression in the *rpoA* K271E *crp*^+^ strain, we investigated the effect of combining AraC mutations in the CTD (R251H) and in the NTD (S14L, in the N-terminal arm, or A152V, in the NTD core). We found that in the *rpoA* N268T *crp*^+^ strain, the combined effect of both AraC mutations was synergistic and not additive (Figure 10D,E). This synergistic effect was most striking in an *rpoA* K271I Δ*crp* strain, reaching about 40% of the expression observed in the *araC*^+^ *rpoA*^+^ *crp*^+^ strain. These results suggest that the R251H mutation in the DNA-binding domain of AraC exerts its effect at a different step of transcriptional activation than mutations in the AraC-NTD, underlying a regulatory mechanism not directly related to AraC-NTD activation.

In conclusion, we have characterized mutations in AraC that functionally compensate for the deleterious effect of *rpoA* mutations. Mutations in the AraC-NTD can almost fully compensate for the absence of Crp. Finally, combinations of mutations in the AraC-NTD and in the AraC-CTD synergistically activate transcription at the P_BAD_ promoter.

## 4. Discussion

The RpoA α-CTD domain is a pleiotropic hub required for transcription activation in multiple systems [8,12]. The residues that functionally interact with each activator are often different. For instance, the E261K/G substitutions affect a subset of Crp-dependent promoters, while P322S affects OmpR [44], and D250 and R310 affect TyrR [45]. The α-CTD has also been shown to play a role in genome-wide transcription regulation in *B. subtilis* [46]. 

Considering the number of *rpoA* mutations detected through our genetic selections, a large fraction of the RpoA residues necessary for P_BAD_ activation have probably now been identified. However, truncations of the α-CTD and several substitutions, including R265A and N268A, are not viable in the absence of a functional RpoA and must be studied in strains containing an *rpoA*^+^ gene on the chromosome [15,47,48]. Thus, our experimental setup does not allow the detection of some mutations, including nonsense ones that would truncate RpoA.

The RpoA K271 substitutions affect CysB and MelR activity in addition to AraC [26]. However, several RpoA C-terminal truncations allow MelR transcriptional activity without any effect on AraC or CysB [11]. More recently, K271 was shown to mediate transcription activation by Fis at the *proP* P2 and *rrnB* P1 promoters, and the Fis residues that interact with the α-CTD were identified through structure-based models [19,49]. The K271E substitution also impaired phage λ PRE activation by *cII* [50,51]. Finally, we observed a 3- to 5-fold decrease in MalE synthesis in strains carrying the *rpoA* K271E substitution; it had no effect on *malPQ* transcription, which is independent of Crp [26]. The A272E change described here has similar effects to the K271E substitution. In contrast, the A272T substitution decreased *ompF* transcription mediated by OmpR, but had no detectable effect on activation by AraC, CysB, and MelR [52]. The replacement of the wild-type A residue by a larger, charged one likely confers a more drastic effect on α-CTD-mediated transcription.

The properties of the *rpoA* mutants provided the basis for the isolation of suppressor mutations that could compensate for the RpoA defects. A similar approach identified mutations in the phage P2 *Ogr* activator gene that suppress the *rpoA109* (L290H) mutation [53]. Furthermore, extensive genetic interactions between TyrR and RpoA have recently been described [45]. In our initial experiments, we used Δ*crp* strains, and only two elements were targets of mutagenesis: the *araC* coding region and the regulatory sequence between *araC* and P_BAD_. Since expression is strongly decreased in the absence of Crp (Figure 3), we used a highly sensitive reporter for screening on indicator plates. In our study, suppressor mutations in genes expressed from the chromosome, such as potential compensatory mutations in *rpoA* or *crp*, could not be identified. The collection of suppressors described here is limited for two reasons. Firstly, the number of characterized mutants is not large, although most *araC* mutations were isolated several times. Secondly, the suppressors were isolated after hydroxylamine treatment of plasmid DNA, using a mutagen that induces only GC to AT transitions, or after mutagenic PCR, which mostly induces both transitions. 

We first isolated two types of suppressors. Mutations in the P_BAD_ promoter are believed to improve intrinsic promoter efficiency, thus reducing the need for activation by AraC (Figure 4). This compensates indirectly for the negative effect of the *rpoA* mutations. The other suppressors carry a mutation in either the N-terminal arabinose-binding and dimerization domain of AraC or the C-terminal DNA-binding domain (Figure 11. Our results suggest that the AraC-NTD is a major target for functional suppression of the RpoA α-CTD mutations and contributes to transcriptional activation of the P_BAD_ promoter, particularly in the absence of Crp. 

The *araC*-NTD mutations largely compensate for the activation defect caused by the absence of Crp in *rpoA*^+^ strains (Figure 5, Figure 6, Figure 7 and Figure 8), thus strengthening activation by AraC. If one assumes that one α-CTD plays a crucial role by interacting with the DNA and possibly with the AraC monomer bound to *I1*, the second α-CTD could bind to another AraC monomer (Figure 1). In the presence of Crp, only one α-CTD would be available to interact with AraC and/or the DNA. The toxic PAI2::PhoA protein confers an Ara^S^ phenotype in an *rpoA*^+^ *crp*^+^ strain but an Ara^R^ phenotype in strains carrying an *rpoA* α-CTD mutation; several *araC* mutations restore the Ara^S^ phenotype in an *rpoA crp*^+^ strain. Thus, a functional interaction of one α-CTD with AraC could also occur in the presence of Crp. 

The positions of the *araC* mutations on the structure of the AraC N-terminal domain [54] are shown in Figure 11A. None of the affected residues are directly involved in arabinose binding. Arabinose constitutive mutations have been isolated in the N-terminal domain of *araC* [41,42,55]. The S14L, N16S, G22D, and A152V mutations confer a constitutive phenotype, with A152V being the most efficient. In contrast, D7N and I71V are devoid of activity in the absence of arabinose but allow efficient induction (Figure 5 and Figure 8). Thus, an arabinose constitutive phenotype is not required for *rpoA* suppression.

Several residues identified here must affect the conformation of the AraC-NTD. I51 and I71 are in the core of the N-terminal domain, on the side of the β-sheet that faces the two α-helices promoting dimerization, with A152 being part of one of these helices. The other residues (D7, S14, and N16) are in the arm of the AraC-NTD, and their lateral chains face the outside of the N-terminal domain. Although it is possible that these residues contact the α-CTD, the major argument against a direct interaction of the flexible arm with the α-CTD lies in the absence of allele specificity between pairs of *araC* and *rpoA* mutant alleles. Therefore, it appears more likely that the *araC* mutations exert an indirect effect and generate a conformation that improves the functional interaction of the α-CTD with the AraC-NTD. 

Alternatively, the AraC-NTD could interact with the RNA polymerase core. If this were the case, the activity of the AraC mutant proteins would be maintained in the presence of an RpoA protein lacking its C-terminal domain. Since this deletion is not viable, we expressed the RpoA deletion protein in strains carrying the *rpoA*^+^ gene on the chromosome. Under these conditions, the activity of wild-type and mutant AraC proteins was inhibited to the same extent (Figure 9 and Appendix A). These results indicate that a putative interaction of the AraC-NTD with the RNA polymerase core does not participate in the suppression of the RpoA defects by the mutant AraC proteins. In agreement with this notion, we did not isolate RNA polymerase mutants, outside of the α-CTD, that selectively decrease P_BAD_ transcription mediated by mutant AraC.

The suppressors that localize to the AraC-NTD suggest a new function for this domain. In the absence of arabinose, the arm domain binds to the DNA-binding domain to ensure DNA looping and P_BAD_ repression. In the presence of arabinose, the arm domain folds on top of the inducer binding pocket, and one of the arm residues, F15, contacts arabinose [41]. In a third function, the AraC-NTD could functionally interact with the RpoA α-CTD to promote efficient transcription at the P_BAD_ promoter, particularly in the absence of Crp. This genetic approach may be easily extended to other activators that interact with the RpoA C-terminal domain, a hub for transcriptional activation.

The first evidence that the α-CTD domain binds DNA was observed with the UP element, upstream of the *rrnB* promoter [15]. R265 is a critical residue for DNA binding, along with several other residues, including N268 [14,20,48]. A crystal structure of a DNA-Crp-α-CTD complex showed that R265 and N268, together with other RpoA residues, bind to the DNA backbone and to the minor groove [14]. The RpoA N268D substitution decreased *lacZ* expression [56], while the N268A, which cannot complement an *rpoA ts* allele [48], and W substitutions abolished activation by an UP element [57]. The R265A and N268A substitutions decreased *rhaBAD* activation by RhaS [58]. The N268T substitution described here abolishes P_BAD_ promoter activity, both in the presence and absence of Crp (Figure 2B and Figure 3B). Unlike the K271 and A272 substitutions, this viable mutation affects at least one more gene than *cysA*, as it prevented growth on minimal media in the presence of Cys and Met.

Transcription activation at P_BAD_ has been proposed to involve the interaction of one of the α-CTD with Crp, and it has been suggested that the other α-CTD could bind to the DNA-binding domain of AraC and to a DNA sequence element between the Crp and AraC binding sites [3] (Figure 1). We did not identify any mutation in the regulatory region that would have defined a DNA site responsible for α-CTD binding, possibly because the target size is small, because a single base pair change would not compensate for the drastic effect of the RpoA N268T change or because it could interfere with AraC binding to I1. Nevertheless, the effect of this change and the role of this residue in DNA binding at other promoters indicate that an α-CTD–DNA interaction could be critical for transcription at the P_BAD_ promoter. The isolation of mutants in the C-terminal DNA-binding domain of AraC that suppress the RpoA N268T allele provides the first genetic evidence to supports this model (Figure 11B). The activity of the R251H mutant in a *rpoA*^+^ Δ*crp* strain strongly suggests that the AraC-CTD could functionally interact with one α-CTD. The synergistic effect of mutations in the NTD and CTD of AraC is compatible with the notion that each domain could function independently to activate transcription.

**Figure 11 microorganisms-12-01928-f011:**
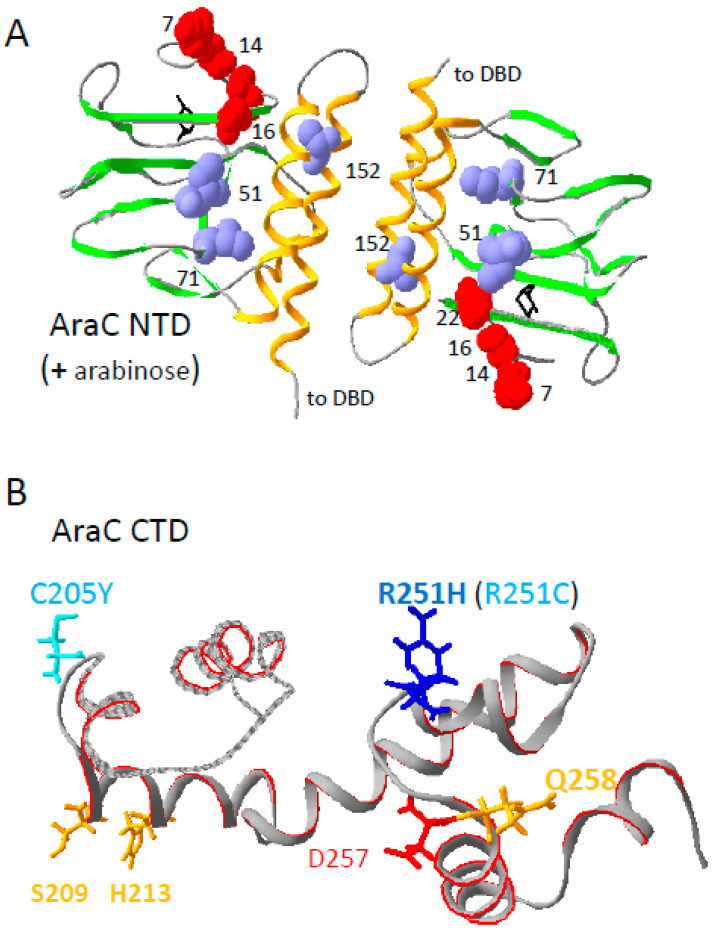
Distribution of the AraC suppressors on the structures of the AraC N-terminal domain dimer (PDB 2ARC) [54] and of the C-terminal DNA-binding domain (PDB 2K9S) [59]. The images were generated with Swiss-PdbViewer v4.1.0. The domains are represented as ribbons. (**A**) The two large α helices that promote NTD dimerization are shown in orange and the β strands are in green. N-terminal arm D7, S14, N16, and G22 residues are shown in red and space-filling. The side chains of the internal residues I51, I71, and A152 are shown in blue and space-filling. The bound arabinose inducers are shown in black. (**B**) The two residues mutated in suppressors of *rpoA* N268T are shown in blue; R251H has the strongest phenotype and is shown in dark blue. Mutations of D257 (red) abolish DNA binding while mutations of the three other residues (orange) are defective in DNA binding.

## Figures and Tables

**Figure 1 microorganisms-12-01928-f001:**
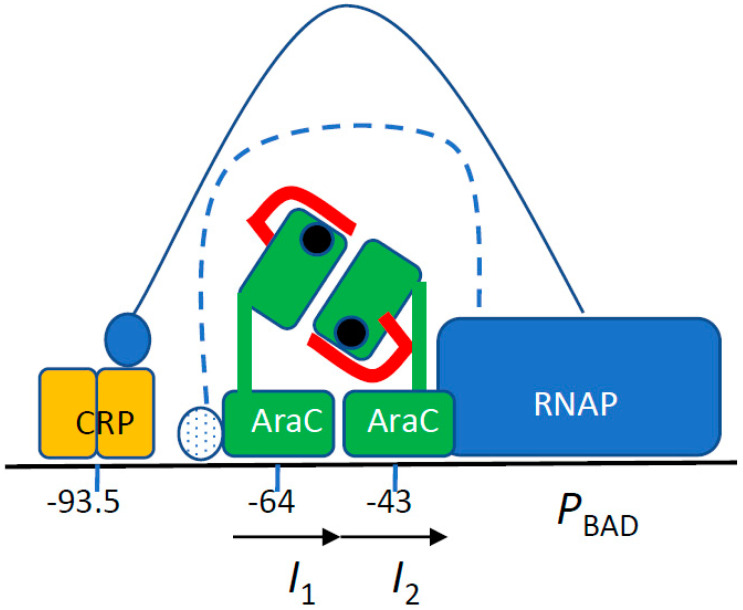
Transcription activation at the arabinose P_BAD_ promoter (adapted from [3]). The sequence represented extends from −120 to +10; the *araC* promoter and *O2* site are not included. RNA polymerase (blue) binding to the promoter is helped by the proximal AraC monomer bound to the *I2* site (position −35 to −51 from the start site) that overlaps the promoter −35 site. Each AraC monomer (green) consists of an N-terminal dimerization domain that binds arabinose (black circle) attached by a linker to the C-terminal DNA-binding domain. In the presence of arabinose, the AraC N-terminal arm (red) covers the sugar-binding pocket. The Crp dimer (yellow) binds to a distal site (position −83 to −104). One α-CTD domain of RpoA contacts the Crp domain (filled blue circle, solid line). A potential contact site for the other α-CTD domain (dotted blue circle, stippled blue line) is indicated.

**Figure 2 microorganisms-12-01928-f002:**
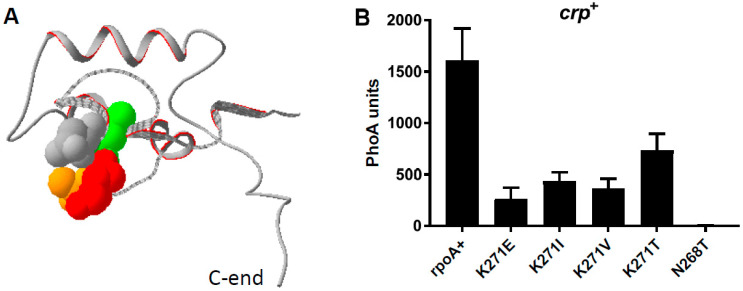
(**A**) The structure of the RpoA C-terminal domain as determined by NMR (PDB 1C00) [17]. The image was generated with Swiss-PdbViewer v4.1.0. The domain is represented as a ribbon, and the side chains of relevant residues are shown in space filling (residues 268–272). N268 is shown in grey, L270 in green, K271 in red and A272 in orange. (**B**) *crp*^+^ strains with the indicated *rpoA* alleles contain the psM74c plasmid expressing wild-type AraC and a Maspin::PhoA chimera. Cultures were induced for 1 h with arabinose.

**Figure 3 microorganisms-12-01928-f003:**
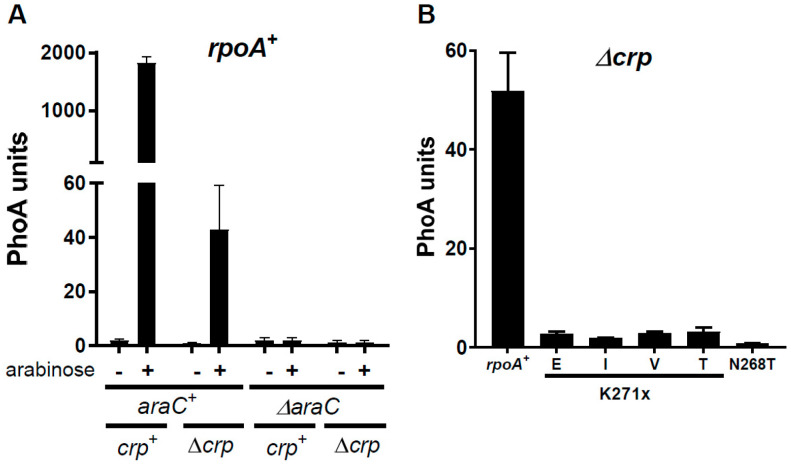
Residual activity of the P_BAD_ promoter in the absence of Crp. (**A**) Plasmid psM74c or its derivative with a complete deletion of *araC* was introduced in *rpoA*^+^ *crp*^+^ and *rpoA*^+^ Δ*crp* strains. Cultures were assayed with or without arabinose induction. (**B**) The psM74c plasmid expressing wild-type AraC was introduced in Δ*crp* strains carrying the indicated *rpoA* alleles. Cultures were induced for 1 h with arabinose.

**Figure 4 microorganisms-12-01928-f004:**
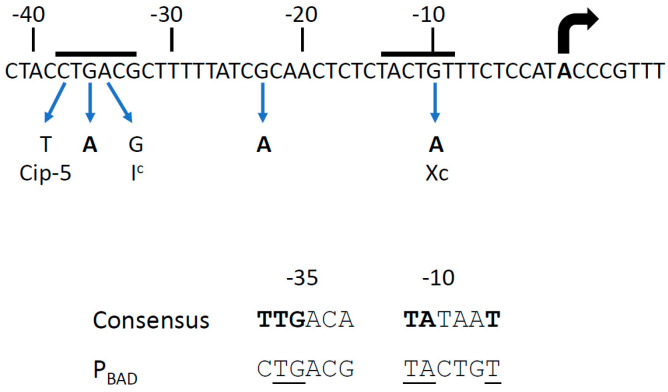
Nucleotide sequence of the P_BAD_ promoter. The positions of the −35 and −10 hexamers and the transcription start site are indicated. The three G to A mutations isolated in this study are shown in bold. Previously described promoter mutations are discussed in the text. The sequence consensus of the −35 and −10 hexamers [40] is compared to that of the P_BAD_ promoter. The most conserved residues in the consensus are in bold, and the conserved residues in the P_BAD_ promoter are underlined.

**Figure 5 microorganisms-12-01928-f005:**
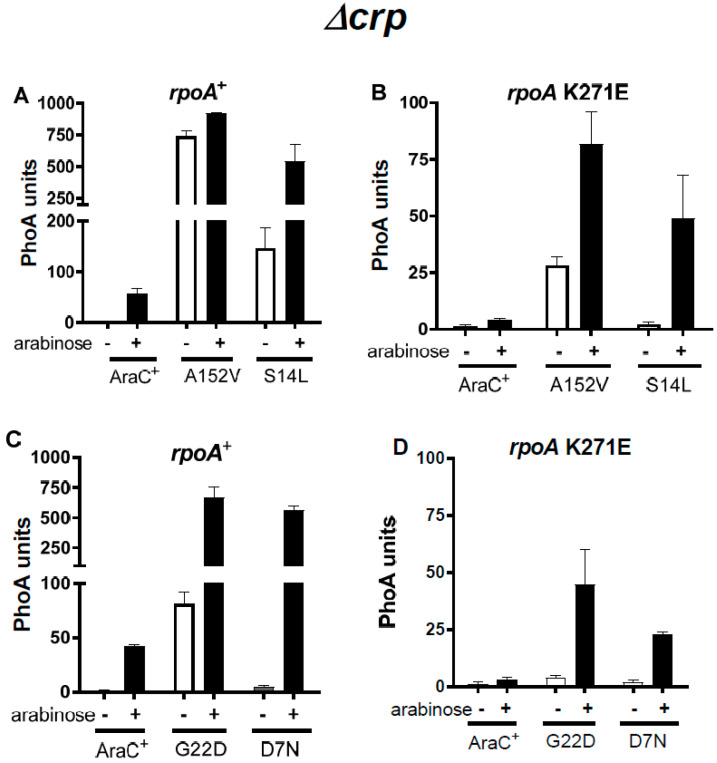
Constitutive P_BAD_ activity mediated by mutant AraC proteins in the absence of Crp. Plasmids expressing AraC^+^ or the indicated AraC suppressor proteins were introduced in *rpoA*^+^ Δ*crp* (**A**,**C**) or *rpoA* K271E Δ*crp* (**B**,**D**) strains. Cultures grown in the absence of arabinose were assayed to detect the constitutive activity of each AraC protein. Cultures induced for 1 h with arabinose were used to determine the full activity of each AraC protein.

**Figure 6 microorganisms-12-01928-f006:**
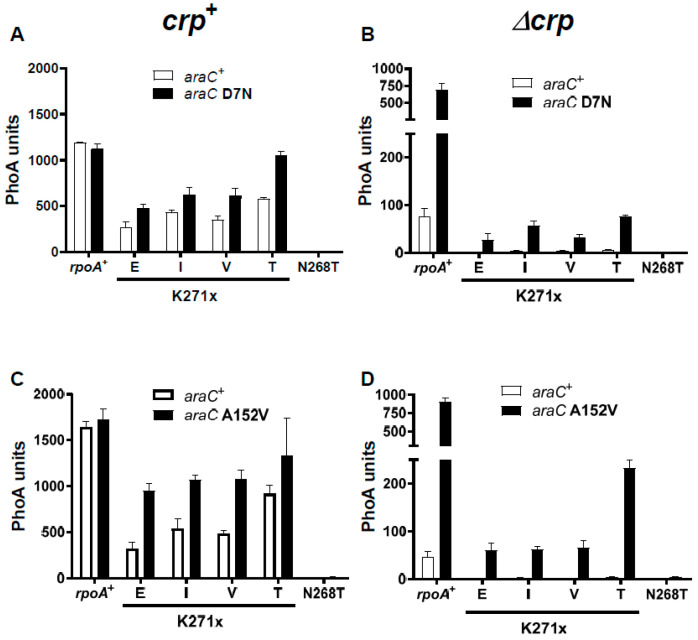
P_BAD_ activity mediated by wild-type AraC^+^ and D7N (**A**,**B**), or A152V (**C**,**D**) mutant AraC proteins. Plasmids expressing AraC^+^ or the indicated AraC mutant proteins were introduced in *crp*^+^ (**A**,**C**) or Δ*crp* (**B**,**D**) strains with the indicated *rpoA* alleles. Cultures were assayed after a 1 h induction with arabinose.

**Figure 7 microorganisms-12-01928-f007:**
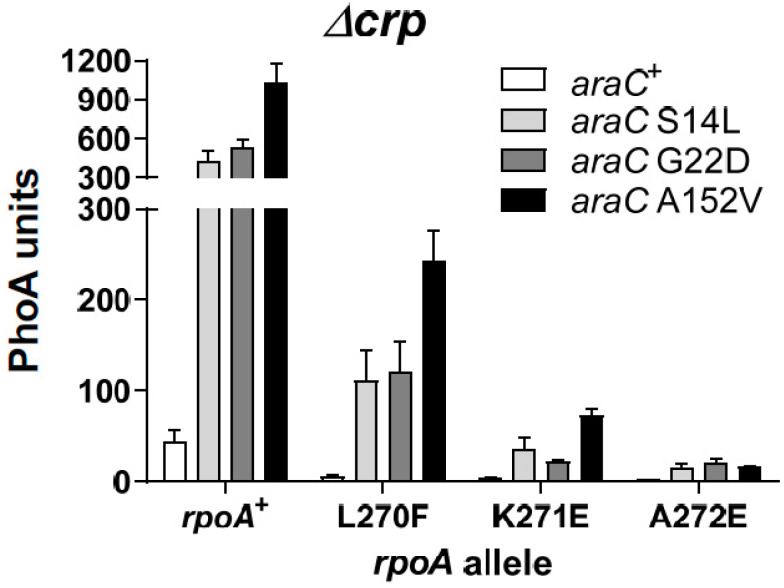
Effects of the AraC suppressors on the two residues that flank K271 of the RpoA α-CTD. Plasmids expressing AraC^+^ or the indicated AraC mutant proteins were introduced in Δ*crp* strains carrying the *rpoA*^+^ or the three indicated *rpoA* mutant alleles.

**Figure 8 microorganisms-12-01928-f008:**
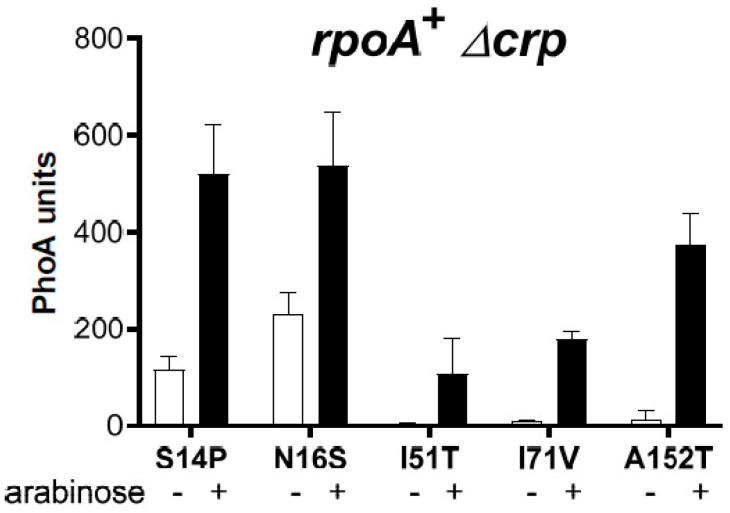
P_BAD_ activity mediated by *araC* suppressors of RpoA A272E. Plasmids expressing AraC^+^ or the indicated AraC suppressor proteins were introduced in the *rpoA*^+^ Δ*crp* strain. Cultures grown in the absence of arabinose were assayed to detect the constitutive activity of each AraC protein. Cultures induced for 1 h with arabinose were used to determine the full activity of each AraC protein.

**Figure 9 microorganisms-12-01928-f009:**
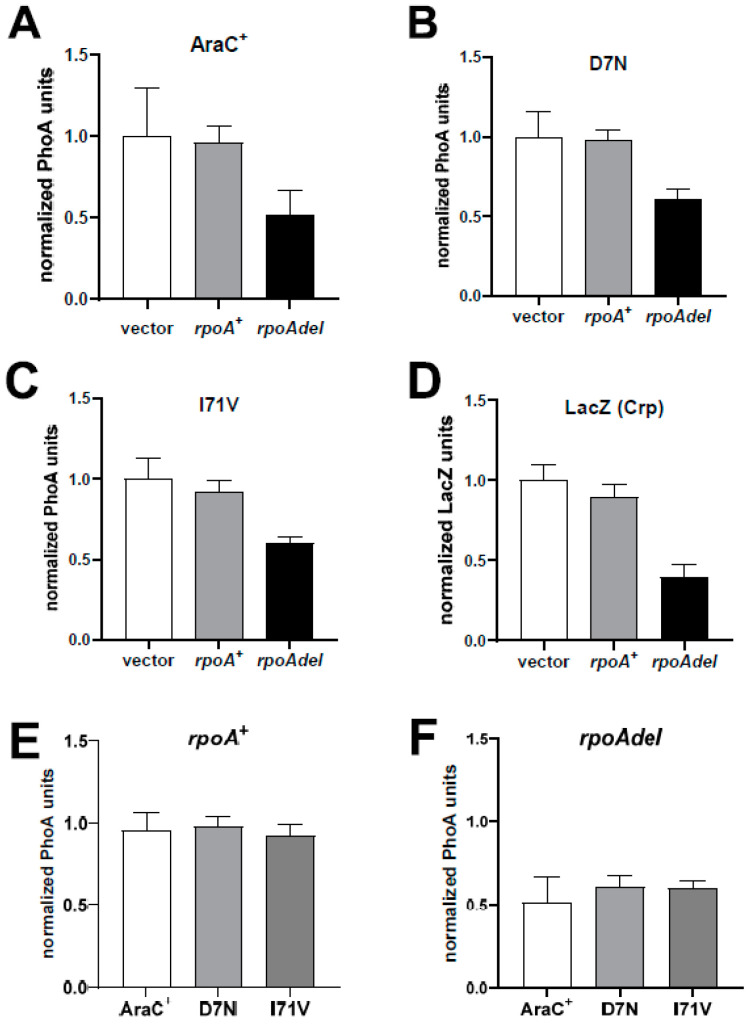
A deletion of the RpoA α-CTD decreases expression from the *araBAD* and *lac* promoters. (**A**–**D**) Empty bars: pUC19 vector; grey bars: pUC-*rpoA*^+^; black bars: pUC-rpoAdelCTD. The PhoA and LacZ activities were normalized to those measured with the empty vector. PhoA expression was measured in *rpoA*^+^ Δ*crp* strains. PhoA units with the empty vector: 4.5 ± 1 (AraC^+^); 390 ± 60 (D7N); 188 ± 30 (I71V). Expression of *lacZ* was measured in the HfrH strain: LacZ units with the empty vector: 3530 ± 340. The normalized activities of the three AraC proteins in the presence of the pUC-*rpoA*^+^ (**E**) or the pUC-rpoAdelCTD (**F**) plasmid are directly compared.

**Figure 10 microorganisms-12-01928-f010:**
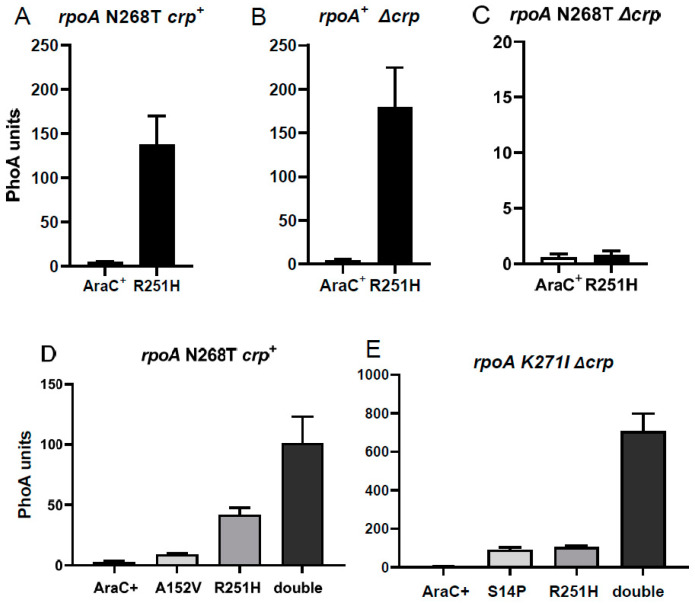
P_BAD_ activity mediated by AraC R251H and AraC double mutants. Plasmids expressing AraC^+^ or the AraC R251H mutant were introduced into the *rpoA* N268T *crp*^+^ (**A**), *rpoA*^+^ Δ*crp* (**B**), or *rpoA* N268T Δ*crp* (**C**) strains. Plasmids expressing AraC^+^, AraC A152V, AraC R251H, or the double mutant A152V-R251H were introduced into the *rpoA* N268T *crp*^+^ strain (**D**). Plasmids expressing AraC^+^, AraC S14P, AraC R251H or the double mutant S14P-R251H were introduced into the *rpoA* K271I Δ*crp* strain (**E**). Cultures were assayed after a 1 h induction with arabinose.

**Table 1 microorganisms-12-01928-t001:** *ropA* mutations that decrease expression from the P_BAD_ promoter.

	*rpoA* Substitutions						
Toxic Protein	N268T	L270F	K271E	K271I	K271V	K271T	A272E
T4 *vs.1*		2	2			2	
PAI2::PhoA			7	3	2	1	2
T4 *55.2*	1						
growth phenotypes							
MC-melibiose	W	R	W	K	K	K	W
min-glucose	no	yes	no	no	no	no	no
+cys-Met	no	yes	yes	yes	yes	yes	yes

## Data Availability

The original contributions presented in the study are included in the article/Appendix A, further inquiries can be directed to the corresponding author/s.

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
