# Peer review of "AraC Functional Suppressors of Mutations in the C-Terminal Domain of the RpoA Subunit of the Escherichia coli RNA Polymerase"

_microorganisms, 2024, doi:10.3390/microorganisms12091928_

Round 1

Reviewer 1 Report

Comments and Suggestions for Authors

In this manuscript titled “AraC functional suppressors of mutations in the C-terminal domain of the RpoA subunit of the Escherichia coli RNA polymerase”, the authors identified mutations clustered in the CTD of the α-subunit of the E. coli RNA polymerase (α-CTD) that downregulate the gene expression from the arabinose PBAD promoter, as well as mutations in the DNA regulatory region and the gene of transcription regulator AraC that upregulate the PBAD expression and thus suppresses the effects of mutations in α-CTD. The following concerns should be addressed before the study is published.

Major concerns:
1. Table 1 is missing in the manuscript.

2. The authors demonstrated opposite effects of mutations in α-CTD and mutations in araC on the PBAD expression. While the araC mutations were identified through recovery of PBAD expression prevented by α-CTD mutations, the argument “The activity of the AraC mutant proteins requires the α-CTD of RpoA” lacks experimental supports. To test the hypothesis that the effect of AraC mutations is dependent on α-CTD, the authors performed experiments with wildtype RpoA (rpoA+) or an RpoA deletion that lacks the α-CTD (rpoAdel) and the results were shown in Figure 9. However, the y-axes (AraC activity/ PBAD expression level) for all figures in Figure 9 are normalized, which makes comparison between data in different figures not possible. To determine whether the effects of araC mutations is dependent or independent of α-CTD, expression levels for wildtype and mutant AraC with RpoA deletion need to be compared. Three new figures from the data in Figure 9 can provide clarification: (1) rpoAdel, x-axis: wildtype AraC and AraC mutants, y-axis: PBAD expression level. (2) rpoA+, x-axis: wildtype AraC and AraC mutants, y-axis: PBAD expression level. (3) vector, x-axis: wildtype AraC and AraC mutants, y-axis: PBAD expression level. If the effect of araC mutations is dependent on α-CTD, the difference of PBAD expression levels among AraC species (WT vs. mutants) should be less for rpoAdel compared to rpoA+.

Author Response

Reviewer 1 & response

Major concerns:
1. Table 1 is missing in the manuscript.

  1. ANSWER The Table was included in the BelinSilvaCostafrolaz figures.pdf file (last page, page 12). The file has been renamed BelinSilvaCostafrolaz figures-table new .pdf for clarity.
  2. The authors demonstrated opposite effects of mutations in α-CTD and mutations in araC on the PBAD expression. While the araC mutations were identified through recovery of PBAD expression prevented by α-CTD mutations, the argument “The activity of the AraC mutant proteins requires the α-CTD of RpoA” lacks experimental supports. To test the hypothesis that the effect of AraC mutations is dependent on α-CTD, the authors performed experiments with wildtype RpoA (rpoA+) or an RpoA deletion that lacks the α-CTD (rpoAdel) and the results were shown in Figure 9. However, the y-axes (AraC activity/ PBAD expression level) for all figures in Figure 9 are normalized, which makes comparison between data in different figures not possible. To determine whether the effects of araC mutations is dependent or independent of α-CTD, expression levels for wildtype and mutant AraC with RpoA deletion need to be compared. Three new figures from the data in Figure 9 can provide clarification: (1) rpoAdel, x-axis: wildtype AraC and AraC mutants, y-axis: PBAD expression level. (2) rpoA+, x-axis: wildtype AraC and AraC mutants, y-axis: PBAD expression level. (3) vector, x-axis: wildtype AraC and AraC mutants, y-axis:. If the effect of araC mutations is dependent on α-CTD, the difference of PBAD expression levels among AraC species (WT vs. mutants) should be less for rpoAdel compared to rpoA+.

    2. ANSWER Our conclusion that “The activity of the AraC mutant proteins requires the α-CTD of RpoA” is based on two experiments with plasmids expressing either rpoA+ or rpoAdel (Figure 9 and supplementary Figure 4). We have reanalysed the data according to the interesting suggestion of the referee and have included two new panels (E & F) in figure 9. Since the experimental values for AraC+ are so different from those with the two AraC mutants, it seems better to normalize the values obtained with the two rpoA plasmids to that obtained with the vector. The PBAD expression levels (PhoA units) with the vector are indicated in the legend of fig. 9. We believe that the new panels provide more visual evidence that the RpoA deletion has the same effect on the activities of the wild type and mutant AraC proteins. Since the three AraC proteins have a similar requirement for the α-CTD, the AraC mutations do not bypass this requirement.

Reviewer 2 Report

Comments and Suggestions for Authors

1. An index for the abbreviation is recommended for easy understanding.

2. The purpose and rationale of the study are suggested to be clarified carefully.

3. The authors resolved the different effects of the obtained mutations on RNA polymerase in this study. What innovative findings are discovered, compared to the reported works?

4. What definite conclusion can be drawn according to the mutation results, not the deduced conclusions?

5. The procedure for site mutation should be clarified. Is it rational design or random mutagenesis? If the rational design was conducted, what are the rules for the specific substitutions?

Comments on the Quality of English Language

The manuscript could be benefit from language editing.

Author Response

Reviewer 2

Comments and Suggestions for Authors

  1. An index for the abbreviation is recommended for easy understanding.
    ANSWER An index for the abbreviations is included on page 2, below the abstract.
  2. The purpose and rationale of the study are suggested to be clarified carefully.
    ANSWER : We have clarified the purpose and rationale of the study in the last paragraph of the Introduction on page 4.
  3. The authors resolved the different effects of the obtained mutations on RNA polymerase in this study. What innovative findings are discovered, compared to the reported works?ANSWER : The rpoA K271E mutation was first described by Giffard & Booth in 1984 (ref. 28) and further characterized by Thomas & Glass (ref.34). The other rpoA mutations described in our manuscript are new alleles as indicated on p. 8, 2nd par. Their differential effects on PBAD as well as on the MelAB and CysA promoters are described in Table 1. The N268T and A272E are new substitutions of previously mutated RpoA residues. They are clearly different from the previously described substitutions since A272T affects other promoters (p. 15, last 3 lines); N268A is not viable since it cannot complement a rpoA ts allele (ref. 47), as mentioned on p. 19, 3rd line.
  4. What definite conclusion can be drawn according to the mutation results, not the deduced conclusions?
    ANSWER : The most definite conclusion on the araC mutations is that they can improve PBAD transcription in strains carrying the indicated rpoA alleles, particularly in the absence of Crp. This is now explicitly stated in the Abstract (line 6). Furthermore, mutations in the N-terminal domain of AraC have a distinct pattern of suppression than mutations in the C-terminal DNA binding domain. This is stated in the Abstract (last two lines). The conclusions are now summarized in the last par. of the results, p. 14.
  5. The procedure for site mutation should be clarified. Is it rational design or random mutagenesis? If the rational design was conducted, what are the rules for the specific substitutions?
    ANSWER : All mutations described in our manuscript result from random mutagenesis. This is now mentioned on pages 2,4,7 & 8.

Comments on the Quality of English Language

The manuscript could be benefit from language editing.

ANSWER : The language was corrected with the help of an English-speaking colleague. The changes are marked in red in the “BelinSilvaJC revised final with marked changes.docx” fil

Round 2

Reviewer 1 Report

Comments and Suggestions for Authors

The authors have adequately addressed all of my concerns, and the study can be published.